# Novel Biological Therapies for Severe Asthma Endotypes

**DOI:** 10.3390/biomedicines10051064

**Published:** 2022-05-04

**Authors:** Corrado Pelaia, Giulia Pelaia, Claudia Crimi, Angelantonio Maglio, Anna Agnese Stanziola, Cecilia Calabrese, Rosa Terracciano, Federico Longhini, Alessandro Vatrella

**Affiliations:** 1Department of Health Sciences, University “Magna Græcia” of Catanzaro, 88100 Catanzaro, Italy; giulia.pelaia@gmail.com; 2Department of Clinical and Experimental Medicine, University of Catania, 95123 Catania, Italy; dott.claudiacrimi@gmail.com; 3Department of Medicine, Surgery and Dentistry, University of Salerno, 84084 Salerno, Italy; amaglio@unisa.it (A.M.); avatrella@unisa.it (A.V.); 4First Division of Pneumology, High Speciality Hospital “V. Monaldi” and University “Federico II” of Naples, Medical School, 80131 Naples, Italy; annastanziola@libero.it; 5Department of Translational Medical Sciences, University of Campania “Luigi Vanvitelli”, 80131 Naples, Italy; cecilia.calabrese@unicampania.it; 6Department of Experimental and Clinical Medicine, University “Magna Græcia” of Catanzaro, 88100 Catanzaro, Italy; terracciano@unicz.it; 7Department of Medical and Surgical Sciences, University “Magna Græcia” of Catanzaro, 88100 Catanzaro, Italy; flonghini@unicz.it

**Keywords:** type 2 severe asthma, monoclonal antibodies, IgE, pro-inflammatory cytokines, alarmins

## Abstract

Severe asthma comprises several heterogeneous phenotypes, underpinned by complex pathomechanisms known as endotypes. The latter are driven by intercellular networks mediated by molecular components which can be targeted by specific monoclonal antibodies. With regard to the biological treatments of either allergic or non-allergic eosinophilic type 2 asthma, currently available antibodies are directed against immunoglobulins E (IgE), interleukin-5 (IL-5) and its receptor, the receptors of interleukins-4 (IL-4) and 13 (IL-13), as well as thymic stromal lymphopoietin (TSLP) and other alarmins. Among these therapeutic strategies, the best choice should be made according to the phenotypic/endotypic features of each patient with severe asthma, who can thus respond with significant clinical and functional improvements. Conversely, very poor options so far characterize the experimental pipelines referring to the perspective biological management of non-type 2 severe asthma, which thereby needs to be the focus of future thorough research.

## 1. Introduction

Asthma is a chronic airway disease affecting more than 300 million people worldwide [1,2]. Such a widespread heterogeneous syndrome is often characterized by reversible airflow limitation associated with bronchial hyperresponsiveness, caused by airway inflammation and remodeling [3]. Asthma heterogeneity is expressed in many clinical phenotypes, including mild, moderate and severe forms, as well as early-onset and late-onset, allergic and non-allergic variants [4]. These phenotypes are driven by complex biologic pathomechanisms named endotypes, consisting of cellular and molecular pathways leading to eosinophilic, neutrophilic, mixed, or paucigranulocytic traits [5,6,7,8,9,10,11].

The above endotypes also characterize severe asthma [12], which is defined as a clinical condition whose control requires high doses of inhaled corticosteroids (ICS) plus long-acting β_2_-adrenergic agonists (LABA), eventually integrated with other drugs including long-acting muscarinic antagonists (LAMA), leukotriene modifiers, theophylline, oral corticosteroids (OCS), and/or targeted monoclonal antibodies [13,14]. The majority of allergic and non-allergic patients with severe asthma present a predominant eosinophilic airway inflammation [15,16]. Indeed, a recent real-world study carried out within the context of the International Severe Asthma Registry (ISAR) suggests that eosinophilic asthma can be detected in more than 80% of patients expressing the most severe disease phenotypes [17,18].

The development of airway eosinophilia depends on pathobiologic networks involving close interactions between innate and adaptive immune responses, occurring in type 2 asthma under the coordination of group 2 innate lymphoid cells (ILC2) and T helper 2 (Th2) lymphocytes, which produce interleukin-5 (IL-5), -4 (IL-4) and -13 (IL-13) [9,10,19,20,21]. The latter are the master mediators of T2-high asthma onset, persistence, and amplification [22]. In particular, IL-5 is the pivotal cytokine responsible for the maturation, proliferation, survival and activation of eosinophils [23]. IL-4 and IL-13 exert key functions related to the biosynthesis of immunoglobulins E (IgE) and to the recruitment of eosinophils into the airways [21]. Moreover, IL-13 is also implicated in the induction of mucus production, bronchial hyperresponsiveness and airway remodeling [24,25]. The release of IL-4, IL-5 and IL-13 from Th2 and ILC2 cells is stimulated by alarmins secreted by the injured bronchial epithelium, exposed to the damaging actions of allergens, respiratory viruses, bacteria, cigarette smoke and airborne pollutants [26]. Alarmins comprise some innate cytokines such as thymic stromal lymphopoietin (TSLP), interleukin-25 (IL-25) and interleukin-33 (IL-33), which work in type 2 asthma as upstream activators of ILC2 and Th2 cells [27].

The above concepts provide a valid explanation in order to understand the relevant roles played by IgE, IL-5 and its receptor, IL-4/IL-13 receptors, and alarmins/alarmin receptors as valuable targets of currently available and potential future add-on biological treatments of type 2 severe asthma (Figure 1) [28,29,30,31,32,33,34].

Differently from type 2 airway inflammation, non-type 2 severe asthma is mainly characterized by a prevalent neutrophilic pattern, sustained by the activation of Th1/ILC1 cells, especially Th17/ILC3 cells releasing interleukin-17 (IL-17) [35,36,37,38]. Unfortunately, at present patients with non-type 2 severe asthma are penalized by a worrisome shortage of reliable biomarkers and effective biologic therapies [39,40]. Taken together, all these considerations imply that a careful phenotypic/endotypic characterization of severe asthma is essential for clinicians to choose the most appropriate biologic treatment, which should be precisely targeted to the pathologic traits of each patient [41,42,43].

On the basis of the previous discussion, this narrative review will focus on two main topics: (i) the pathobiology of severe asthma; (ii) current and perspective biological therapies of severe asthma.

## 2. Pathobiology of Severe Asthma

Severe asthma is often characterized by a type 2 eosinophilic endotype [15,16,17,18], triggered by either allergic or non-allergic mechanisms. Allergic asthma is initiated by the inhalation of aeroallergens, which are captured by conventional dendritic cells, processed and exposed in thoracic lymph nodes to T cell receptors of naïve CD4^+^ T lymphocytes within the context of HLA class II molecules of the major histocompatibility complex (MHC II), in the presence of specific co-stimulatory molecules (CD80-CD86/CD28, OX40/OX40 ligand, ICOS/ICOS ligand) [1,44]. Such a modality of antigen presentation, associated with a specific cytokine milieu expressed by adequate levels of mast cell-/basophil-derived IL-4, induces the polarization of naïve CD4^+^ Th0 cells towards the Th2 immunophenotype [45]. Non-allergic eosinophilic asthma is mainly orchestrated by ILC2 [46]. Both Th2 and ILC2 cells are activated by epithelial alarmins (TSLP, IL-25, IL-33), which directly stimulate ILC2 to secrete type 2 cytokines (IL-4 and, especially, IL-5 and IL-13) and also solicit dendritic cells to promote Th cell commitment towards the Th2 lineage [26,27,47]. Once matured and activated, Th2 lymphocytes migrate to the airway district, where they release copious amounts of IL-4, IL-5 and IL-13 [20,21]. Together with IL-4, also produced by T follicular helper (Tfh) cells, IL-13 drives Ig class switching thus inducing B lymphocytes to synthesize antigen-specific immunoglobulins E (IgE), which bind to high-affinity (FcεRI) and relatively low-affinity (CD23/FcεRII) receptors, expressed by both immune/inflammatory and structural cells of the airways [20,48].

In addition to lymphocytes, other cellular targets of IL-4 and IL-13 include eosinophils and bronchial epithelial cells. In particular, these cytokines foster the vascular endothelial margination of eosinophils and their migration to the airways [49,50]. IL-4 and IL-13 contribute to asthma pathogenesis also by impairing the integrity of the airway epithelium [51,52]. Furthermore, IL-13 elicits goblet cell hyperplasia and mucus overproduction [53], as well as upregulates bronchial epithelial expression of the inducible isoform of nitric oxide synthase (iNOS), thereby enhancing the airway levels of nitric oxide [54], a very useful biomarker of type 2 asthma. Moreover, IL-13 is involved in the pathobiology of airway remodeling by increasing collagen production and proliferation of both bronchial fibroblasts and airway smooth muscle cells [24,25,55,56].

The biologic effects of IL-4 and IL-13 are mediated by their binding to two heterodimeric receptors, consisting of either the α-subunit of IL-4 receptor (IL-4Rα) associated with the γc chain (type I receptor) or IL-4Rα coupled with the IL-13 receptor α1-subunit (type II receptor) [57,58,59,60]. The interactions of IL-4 and IL-13 with their receptors, located on the surface of target cells, are followed by the engagement of a complex intracellular signaling network, which is based on the JAK (Janus kinases 1)–STAT6 (signal transducer and activator of transcription 6) system responsible for the activation of the transcription factor GATA3 [61,62,63,64,65,66,67]. The latter in turn switches on the expression of the genes encoding IL-4, IL-5 and IL-13, thus nurturing a feed-forward molecular circuit which leads to the persistence and amplification of type 2 inflammation in asthma [66,67]. Differently from IL-13 receptor α1-subunit (IL-13Rα1), IL-13Rα2 binds IL-13 but does not activate any downstream post-receptor pathway, thereby behaving as a negative regulator of IL-13 signaling [61].

IL-5 is the pivotal cytokine implicated in the differentiation, survival, proliferation, activation and degranulation of eosinophils [22,23]. Besides Th2 lymphocytes, eosinophils themselves, mast cells, natural killer cells, and especially ILC2 contribute to the production of IL-5 [23]. In addition to synergistically acting together with Th2 cells in allergic asthma, ILC2 are the most relevant cellular sources of IL-5 in non-allergic eosinophilic asthma [46], whose clinical manifestations often start in adulthood (late-onset asthma). The biologic effects of IL-5 are mediated by its interaction with the IL-5 receptor (IL-5R). This receptor comprises an IL-5-specific α subunit (IL-5Rα) and a non-specific βc chain and can bind which IL-5, IL-3 and granulocyte macrophage-colony stimulating factor (GM-CSF) [68]. IL-5 binding to IL-5Rα drives the assembly of a binary IL-5Rα/βc receptor complex, which triggers the activation of an intricate signal transduction network [69]. The latter includes the JAK2–STAT1/3/5 functional unit leading to eosinophil proliferation, as well as a heterogeneous group of kinases (Raf-1, MAPK, PI3K) which are responsible for several actions of IL-5 such as eosinophil survival, activation, degranulation and inhibition of apoptosis [70,71]. High serum levels of IL-5 are present in severe asthmatic patients, in whom eosinophilopoiesis occurs not only in the bone marrow, but also in the airways [70]. Moreover, IL-5 suppresses eosinophil apoptosis and promotes eosinophil recruitment into the airways [72], thereby synergizing with the powerful eosinophil chemoattractant eotaxin [73]. Furthermore, in patients with type 2 asthma, IL-5 stimulates eosinophils to interact with the extracellular matrix protein periostin, whose levels are upregulated when eosinophils accumulate in the airways [74]. IL-5 also provides a key signal for eosinophil degranulation [75], thus favoring the damage of both airway epithelium and neural tissue through the release of the eosinophil granule content, consisting of cytotoxic proteins which include eosinophil cationic protein, eosinophil peroxidase, major basic protein and eosinophil-derived neurotoxin [76,77]. Another mediator secreted by eosinophils when activated by IL-5 is transforming growth factor-β1 (TGF-β1), which significantly concurs to airway remodeling in severe asthma via its fibrogenic effects [78]. Eosinophils are also able to implement a process called ETosis, based on the generation and release of eosinophilic extracellular traps (EET), formed by web-like structures composed of granule proteins and mitochondrial DNA, which contribute to airway inflammation in severe asthma [79,80].

Non-type 2 severe asthma is often characterized by airway neutrophilia [37]. This inflammatory pattern can be induced by several noxious agents comprising bacterial and viral infections, cigarette smoking, airborne pollutants and occupational chemicals [81]. Such environmental triggers activate innate immune receptors including toll-like receptors (TLRs) and pattern recognition receptors (PRRs), expressed on the surface of both airway epithelial cells and immune/inflammatory cells [82]. TLR stimulation leads to the activation of immune responses operated by Th1 and Th17 lymphocytes [83]. In particular, Th17 cell differentiation requires the coordinated intervention of IL-1β, TGF-β, IL-6 and IL-23 [84,85]. Furthermore, PRR interactions with danger-associated molecular patterns (DAMPs) and pathogen-associated molecular patterns (PAMPs) activate the NLRP3 (nucleotide-binding oligomerization domain-like receptor family, pyrin domain containing 3 activation) inflammasome [86]. The latter consists of an intracellular multiprotein complex responsible for the stimulation of the protease caspase-1, which converts the IL-1β precursor into the biologically active cytokine, crucially engaged in the commitment to the Th17 immunophenotype [87]. This inflammasome pathway is also implicated in the pathophysiology of bronchial hyperresponsiveness associated with obesity [88]. In severe asthma, neutrophilic airway inflammation and Th17 polarization are also promoted through NETosis, a process elicited by either infectious or non-infectious stimuli. NETosis consists in the release of neutrophil traps (NETs), which are scaffold-like structures made of extracellular DNA, histones and granular proteins extruded by neutrophils, thus becoming anuclear elements called cytoplasts [89]. Once differentiated and activated, Th17 lymphocytes play a central role in the induction, persistence and amplification of neutrophilic asthma [35]. Indeed, both Th17 lymphocytes and type 3 innate lymphoid cells (ILC3) secrete IL-17A and IL-17F [35,90], which are overexpressed in bronchial biopsies of some patients with severe asthma [36]. In this regard, it is noteworthy that high ILC3 numbers are present in the bronchoalveolar lavage fluid (BALF) from severe asthmatic subjects [91]. Acting at the level of bronchial epithelial cells, sub-epithelial airway fibroblasts and monocytes/macrophages, IL-17A and IL-17F significantly increase the production of powerful neutrophil chemoattractants such as CXCL8 (IL-8) and CXCL1 (GRO-α) [92,93,94]. IL-17-dependent neutrophilic asthma is often associated with a pharmacologic resistance to corticosteroids, which suppress the apoptotic death of neutrophils and prolong their survival [95]. In addition to Th17 cells, a minor role in severe neutrophilic asthma is also played by IL-12-driven differentiation and activation of Th1 cells [81]. In fact, Th1 cell count and the production of the Th1 cytokine interferon-γ (IFN-γ) appear to be enhanced in severe asthma [81].

The mixed neutrophilic/eosinophilic endotype can be quite frequently detected in severe asthmatic patients. Indeed, Th2/Th17 cell lineages releasing both IL-4 and IL-17A have been found in blood samples obtained from asthmatic subjects [96]. Moreover, dual-positive Th2/Th17 clones secreting copious quantities of IL-4 and IL-17 are present in BALF taken from patients with severe asthma [97]. In particular, these BALF T lymphocytes can be characterized by a concomitant expression of the two transcription factors GATA3 and RORγt [97]; the first one is required for Th2 cell commitment, whereas the second one is essential for Th17 polarization [20,98]. Such observations corroborate the results of previous murine studies, showing the participation of both Th2 and Th17 lymphocytes in the experimental development of severe asthmatic endotypes [99].

Paucigranulocytic asthma is a further phenotype/endotype, apparently characterized by the dissociation of airflow limitation and bronchial inflammation, suggested by the lack of a relevant granulocytic infiltration of airway walls [11]. Therefore, it has been hypothesized that paucigranulocytic asthma is a consequence of structural changes regarding airway smooth muscle cells [11]. However, even in the substantial absence of eosinophils and neutrophils, the involvement of inflammatory mechanisms cannot be definitely ruled out [81]. Indeed, other inflammatory cell types such as mast cells could infiltrate the airway smooth muscle cell layer, thereby releasing mediators implicated in airflow limitation, airway remodeling and bronchial hyperresponsiveness [100].

## 3. Currently Licensed Biological Therapies of Severe Asthma

On the basis of the recommendations reported in step 5 of GINA (Global Initiative for Asthma) guidelines [101], currently available biological therapies for severe asthma include anti-IgE, anti-IL-5, anti-IL-5R and anti-IL-4R add-on treatments (Figure 1). Moreover, with regard to these therapies, the fully human anti-TSLP monoclonal antibody tezepelumab (Figure 1) was approved by the U.S. Food and Drug Administration (FDA) on 17 December 2021.

### 3.1. Anti-IgE Therapy

The humanized anti-IgE monoclonal antibody omalizumab was the first biologic drug approved by regulatory agencies for the add-on treatment of severe asthma [102]. Through a specific binding to the two Cε3 domains of the constant fragment of human IgE, omalizumab induces the assembly of IgE/anti–IgE immune complexes which inhibit IgE interactions with high-affinity FcεRI and low-affinity FcεRII/CD23 receptors [103]. Therefore, omalizumab is very effective in preventing all biological actions exerted by IgE at the level of airway immune/inflammatory and structural cells [104]. Eligible patients for omalizumab include subjects not adequately controlled by high ICS dosages and experiencing frequent asthma exacerbations, with positive skin prick test and/or RAST (radioallergosorbent test) for perennial allergens, serum IgE levels ranging from 30 to 1500 IU/m, and a forced expiratory volume in one second (FEV_1_) less than 80% of the predicted volume [105]. Within the above patient population, the most responsive people to the therapeutic effects of omalizumab are those characterized by high levels of blood eosinophils, serum periostin and fractional exhaled nitric oxide (FeNO) [106].

Several randomized controlled trials (RCTs), performed before and after omalizumab approval, have clearly demonstrated its efficacy in decreasing the annual number of asthma exacerbations in allergic patients [107]. Such positive clinical benefits have been further confirmed by many real-life investigations carried out worldwide [102]. In particular, these real-world studies have clearly shown that omalizumab, in addition to significantly lowering the rate of asthma exacerbations and the accesses to emergency room and hospital wards, reduces OCS intake and the loss of both school and working days [102,108]. Furthermore, real-life experiences convincingly suggest that omalizumab is able to elicit significant and persistent FEV_1_ increases, measurable 5, 7 and even 9 years after starting the anti-IgE therapy [109,110]. The achievement of all these satisfactory clinical and functional results explains the high level of treatment adherence reported by patients undergoing anti-IgE therapy [111]. Moreover, omalizumab is characterized by a long-lasting good profile of safety and tolerability [112].

### 3.2. Anti-IL-5 Treatments

Mepolizumab is a humanized IgG1/κ monoclonal antibody which selectively targets IL-5, thus inhibiting its binding to IL-5Rα [69,113]. Mepolizumab was initially shown to be effective in small numbers of uncontrolled patients with severe eosinophilic asthma, experiencing frequent exacerbations [114,115]. Indeed, in these subjects mepolizumab significantly lowered asthma exacerbations and also decreased blood/sputum eosinophil counts. Such relevant findings were later replicated by the phase 2b/3 DREAM trial, conducted in a much more numerous study population including subjects with severe eosinophilic asthma [116]. In addition, both phase 3 MENSA and SIRIUS studies showed that in patients with severe eosinophilic asthma, mepolizumab not only decreased disease exacerbations, but also significantly improved symptom control and quality of life, as well as slightly incremented FEV_1_ [117,118]. Furthermore, the SIRIUS trial documented that mepolizumab treatment exerted a 50% OCS-sparing effect [118]. Another phase 3b RCT, named MUSCA, further confirmed the beneficial therapeutic role played by mepolizumab with regard to health-related quality of life [119]. Moreover, a recent post hoc meta-analysis of MENSA and MUSCA studies evidenced that mepolizumab effectively ameliorated physical activity level and work productivity in subjects with severe eosinophilic asthma [120]. Patients who had been previously enrolled in either MENSA or SIRIUS RCTs were also recruited in the COSMOS open-label, phase 3b extension study, which convincingly proved the long-term safety and efficacy of mepolizumab [121].

Current real-life data clearly corroborate the above RCT findings referring to the therapeutic action exerted by mepolizumab as add-on biological treatment of severe eosinophilic asthma. Surprisingly, real-world investigations indicate that mepolizumab can be even more efficacious in comparison to RCTs, especially with regard to its capability of reducing the number of clinically significant asthma exacerbations [122]. This difference could be at least in part explained by the occurrence of higher blood eosinophil counts, often exhibited by patients participating in real-life observational studies [123]. Indeed, based on the intensity of clinical responses, in real-world settings it is possible to detect super-responders to mepolizumab, who express very high levels of biomarkers related to type 2 airway inflammation [124]. Real-life experiences also suggest that mepolizumab is equally effective in patients with either allergic or non-allergic severe eosinophilic asthma [125]. In fact, mepolizumab can be a valuable option as a successful switching treatment for allergic subjects with severe eosinophilic asthma, unresponsive to an initial biological therapy with omalizumab [126,127]. In patients with severe eosinophilic asthma, mepolizumab can improve lung function by increasing FEV_1_ and forced expiratory flow over the middle half of vital capacity (FEF_25–75%_) [128], thereby acting at the level of both large and small airways.

Another biologic targeting IL-5 is the humanized IgG4/κ monoclonal antibody reslizumab, which has been tested in many RCTs [129]. Differently from the other biologic drugs used for the treatment of severe asthma, which are administered via the subcutaneous route, reslizumab is injected intravenously. The first phase 2 study demonstrated that reslizumab lowered blood and sputum eosinophil counts, as well as transiently enhanced FEV_1_ [130]. A further phase 2 trial, conducted in patients with severe eosinophilic asthma, showed that reslizumab significantly increased FEV_1_ and also induced a non-significant improvement in symptom control, markedly in people characterized by high blood eosinophil numbers and coexistent nasal polyps [131]. Two other phase 3 studies were later performed in severe asthmatic patients with more than 400 blood eosinophils/μL, whose therapeutic responses to reslizumab comprised a higher than 50% reduction in the annual asthma exacerbation rate, an improvement in asthma control and a relevant FEV_1_ enhancement [132]. These findings were subsequently confirmed, especially in study populations characterized by the presence of patients with late-onset eosinophilic asthma [133]. Furthermore, the beneficial effects on lung function experienced by patients treated with reslizumab not only were expressed by FEV_1_ increments, but also included significant FEF_25–75%_ increases [134]. The positive impact of reslizumab on asthma exacerbations, symptom control and FEV_1_ has also been observed in real-life clinical investigations [135]. Overall, reslizumab presents a good safety and tolerability profile [129]; however, a pooled analysis of 6 clinical trials involving 1028 patients treated with this anti-IL-5 biologic drug allowed to identify 3 successfully managed cases of anaphylaxis [136].

### 3.3. IL-5 Receptor Blockade

In contrast to mepolizumab and reslizumab, the humanized and afucosylated IgG1/κ monoclonal antibody benralizumab occupies and blocks the IL-5 receptor [137]. In particular, the molecular target of the Fab fragments of benralizumab is IL-5Rα, whereas the constant Fc portion interacts with the FcγRIIIa receptor expressed by natural killer (NK) cells, thus driving eosinophil apoptosis through antibody-dependent cell-mediated cytotoxicity (ADCC), a mechanism which is markedly amplified by antibody afucosylation [137]. Therefore, benralizumab exerts is anti-eosinophil actions by inhibiting IL-5 functions at the receptor level, as well as by directly killing eosinophils. Benralizumab has been extensively evaluated in patients with severe eosinophilic asthma by several phase 3 RCTs, including SIROCCO and CALIMA, which showed that this biologic significantly lowered the number of asthma exacerbations and also ameliorated symptom control and pulmonary function [138,139]. The beneficial effect of benralizumab on lung function was further confirmed by the phase 3 BISE trial, which demonstrated that this drug increased FEV_1_ in subjects with eosinophilic asthma and a blood eosinophil count ≥300 cells/μL [140]. More recently, the ANDHI phase 3b trial showed that benralizumab, in addition to decreasing asthma exacerbation rate and increasing FEV_1_ and peak expiratory flow (PEF), improved symptom control and health-related quality of life [141]. Both ZONDA and PONENTE studies clearly documented that benralizumab is able to provide a significant OCS-sparing therapeutic action in patients with severe eosinophilic asthma [142,143]. Moreover, the BORA phase 3 extension trial has shown that benralizumab is characterized by a persistent good profile of safety and tolerability [144]. The results of RCTs and real-world experiences have also shown that benralizumab plays a relevant role as add-on biological therapy in both allergic and non-allergic patients with severe eosinophilic asthma [145,146]. Furthermore, in real-life, benralizumab can be successfully used as switching biologic treatment for subjects with severe eosinophilic allergic asthma, experiencing a weak therapeutic response to omalizumab [147]. Overall, the findings reported by RCTs are currently being corroborated and integrated by real-world observations indicating that in patients with severe eosinophilic asthma, benralizumab is very effective in depleting blood eosinophils, decreasing the disease exacerbation rate and OCS intake, as well as in improving symptom control, airflow limitation and lung hyperinflation [148,149].

### 3.4. IL-4/IL-13 Receptor Blockade

The fully human IgG4 monoclonal antibody dupilumab suppresses the biological actions of both IL-4 and IL-13 by selectively binding to IL-4Rα, shared by these two cytokines for the activation of their receptor mechanisms [64]. Initially, a phase 2a trial and a larger phase 2b study showed that dupilumab significantly lowered asthma exacerbation rate and increased FEV_1_, especially, but not exclusively, in patients with relatively high blood eosinophil counts [150,151]. Subsequently, the key phase 3 LIBERTY ASTHMA QUEST trial confirmed that dupilumab decreased asthma exacerbation number and enhanced FEV_1_ by more than 200 mL [152]. This study also demonstrated that dupilumab transiently incremented blood eosinophil numbers in some patients, whereas significantly reduced other biomarkers of type 2 inflammation, including FeNO and the blood levels of IgE, eotaxin-3, periostin and thymus- and activation-regulated chemokine (TARC) [152]. Moreover, a post-hoc analysis of LIBERTY ASTHMA QUEST reported that dupilumab was effective in both allergic and non-allergic asthmatics [153]. Another phase 3 study was the LIBERTY ASTHMA VENTURE trial, which provided strong evidence about the OCS-sparing effect induced by dupilumab in patients with OCS-dependent severe asthma [154]. More than 2000 asthmatic subjects recruited in the above two studies were also enrolled in a long-lasting (96 weeks), open-label extension trial called TRAVERSE, whose primary goal was to evaluate the long-term safety of dupilumab [155]. According to this study, dupilumab resulted to be quite safe and well tolerated and also elicited a progressive reduction of blood eosinophils and serum IgE [155]. LIBERTY ASTHMA VOYAGE was a further phase 3 trial which recently showed in children with moderate-to-severe asthma that dupilumab was able to significantly decrease the annualized rate of severe asthma exacerbations, as well as to ameliorate symptom control and lung function [156].

Taken together, the above studies convincingly suggest that dupilumab is mostly effective in severe asthmatics having at least 150 eosinophils per microliter of blood and/or at least 25 parts per billion (ppb) of FeNO [152,157], who eventually also undergo long-lasting OCS treatment. When used in such patients within the context of real-life settings, dupilumab exerts valuable therapeutic effects including rapid and relevant improvements in asthma exacerbations, symptom control, airflow limitation, lung hyperinflation and OCS intake [158,159,160]. Dupilumab can be currently prescribed as add-on biological therapy not only for severe asthmatic patients, but also for subjects with either atopic dermatitis or nasal polyposis, which are known to be two very important asthma comorbidities [161,162,163].

### 3.5. Anti-TSLP Therapy

On 17 December 2021, the fully human anti-TSLP monoclonal IgG2λ antibody tezepelumab was approved by the FDA for add-on maintenance therapy of severe asthma, with no phenotype or biomarker limitation [164]. Tezepelumab targets a pivotal mediator involved in the pathobiology of type-2 asthma [30,31]. Indeed, TSLP upregulates OX40 ligand expression on dendritic cells, thereby inducing them to drive naïve Th cell polarization towards the mature and active Th2 immunophenotype, secreting IL-4, IL-13 and IL-5 [165,166]. Additionally, TSLP is a powerful direct activator of ILC2, which respond to this alarmin by prolonging their survival and releasing the above-mentioned type 2 cytokines, as well as by exhibiting corticosteroid resistance [167,168].

Tezepelumab specifically binds to TSLP thus inhibiting its interaction with the TSLP receptor complex and the consequent downstream activation of a signaling network including several pathways operated by the JAK1/2–STAT3/5 transducing system, as well as by phosphoinositide 3 kinase (PI3K), mitogen-activated protein kinases (MAPK) and nuclear factor-κB (NF-κB) [169,170,171,172]. The phase 2b PATHWAY trial showed that tezepelumab reduced by 60–70% the annualized rate of asthma exacerbations and increased pre-bronchodilator FEV_1_, independently of baseline blood eosinophil counts [173]. In addition, tezepelumab decreased the levels of key biomarkers of type 2 asthmatic inflammation such as FeNO, blood eosinophils and serum IgE [173]. Later on, a program of phase 3 studies was developed, named PATHFINDER and including the NAVIGATOR and SOURCE trials [30,31]. The NAVIGATOR study clearly confirmed, in patients with moderate-to-severe asthma, that tezepelumab can act regardless of blood eosinophil counts by decreasing asthma exacerbation rate by 56% and improving symptom control, pulmonary function and health-related quality of life [174]. Once again, this study also reiterated that tezepelumab lowered FeNO, blood eosinophils and serum IgE. Preliminary data provided by the SOURCE trial suggest that tezepelumab was not able to exert a significant OCS-sparing action in the whole study population of enrolled severe asthmatic patients; however, this biologic induced a greater reduction of OCS intake in subjects with high blood eosinophil counts [175].

Outside of the PATHFINDER program, other studies such as DESTINATION and CASCADE have been designed. DESTINATION is a long-lasting extension trial aimed to evaluate in severe asthmatic patients the safety profile of tezepelumab and its prolonged effects on asthma exacerbations [176]. The CASCADE study was conducted in patients with moderate-to-severe asthma undergoing a comprehensive evaluation of cellular inflammatory features, performed through endoscopic tissue sampling of the airways, which showed that tezepelumab significantly reduced the eosinophilic infiltration of the bronchial submucosa, but did not affect the amount of T lymphocytes, mast cells and neutrophils [177]. Furthermore, tezepelumab caused a marked decrease of eosinophil numbers in both BAL and bronchial biopsies obtained from adult asthmatics, as shown by the UPSTREAM trial which also documented a tezepelumab-induced, not significant attenuation of airway hyperresponsiveness to mannitol [178]. Besides promoting type 2 inflammation, TSLP can also favor the commitment of naïve Th lymphocytes towards a Th17 cell lineage [179]. This consideration implies that tezepelumab should be tested for a potential additional efficacy in Th17 cell-driven non-type 2 asthma [180].

## 4. Experimental Biological Therapies of Severe Asthma

IL-33 and IL-25 are two other alarmins that cooperate with TSLP to induce ILC2 activation and Th2 cell polarization [27,181,182]. A recent phase 2 trial showed that the anti-IL-33 monoclonal antibody itepekimab, administered subcutaneously to patients with moderate-to-severe asthma at the dosage of 300 mg every 2 weeks for 12 weeks, improved asthma control, lung function and quality of life, as well as lowered blood eosinophil count [183]; this latter effect was also observed when itepekimab was administered in association with dupilumab. Moreover, itepekimab decreased FeNO, plasma eotaxin-4, serum total IgE and periostin; however, such effects were inferior to those elicited by dupilumab alone or by the itepekimab–dupilumab combination [183]. Another anti-IL-33 antibody is etokimab, whose potential ability to decrease blood eosinophil numbers is currently under evaluation in a phase 2a trial carried out in subjects with severe eosinophilic asthma [30]. Differently from itepekimab and etokimab, astegolimab targets the ST2 (suppression of tumorigenicity 2) receptor of IL-33. The phase 2b ZENYATTA trial showed that subcutaneous injections of 490 mg of astegolimab, repeated every 4 weeks for 52 weeks, significantly and safely decreased the annualized asthma exacerbation rate in adults with severe asthma [184]. Outside of the monoclonal antibody scenario, a fusion protein called IL-33 trap has been engineered, originating from the combination of the extracellular domains of the ST2 receptor with the co-receptor IL-1RAcP (IL-1 receptor accessory protein) [185]. In particular, this molecular construct neutralizes IL-33 and inhibits allergic airway inflammation in pre-clinical experimental models [185].

Otherwise than type 2 airway inflammation (Figure 1), there are only a few perspective options for biological therapies of non-type 2 severe neutrophilic asthma [39,40,186,187]. In this regard, the main targets of monoclonal antibodies undergoing experimental evaluation include the molecular components of the IL-23/IL-17 pathogenic axis. However, a recent phase 2a trial showed that the anti-IL-23 monoclonal antibody risankizumab was not able to reduce sputum neutrophil count and the annualized rate of asthma exacerbations [188]. Utilized in a murine model of ozone-induced airway inflammation, the anti-IL-17A antibody secukinumab decreased neutrophil count and IL-8 levels in BAL fluid [189]. Brodalumab is a fully human monoclonal antibody that targets the IL-17 receptor. Administered to patients with moderate-to-severe asthma, brodalumab did not change asthma symptoms and pulmonary function in the entire study population [190]. Nevertheless, this biologic improved asthma control in a small subgroup of subjects who were characterized by a marked bronchodilator reversibility [190].

## 5. Conclusions

The recent progress in our understanding of the pathomechanisms underlying the heterogeneous endotypes of severe asthma has led to the development, approval and practical use of many biologic drugs including omalizumab, mepolizumab, reslizumab, benralizumab and dupilumab. The efficacy and safety of these antibodies as add-on treatments of type 2 airway inflammation have been convincingly shown by both RCTs and real-life observations. Moreover, the above biologic drugs provide significant benefits also to a very relevant comorbidity of severe asthma such as nasal polyposis [141,191,192,193,194,195,196]. The biomarker-guided choice of currently available biologics for the treatment of severe asthma is reported in Table 1. A further monoclonal antibody recently approved by the FDA for the biological therapy of severe asthma is tezepelumab, whereas other anti-alarmins are under ongoing clinical evaluation. Unfortunately, the great therapeutic opportunities currently enjoyable by subjects with allergic and eosinophilic asthma do not regard patients with predominant neutrophilic bronchial inflammation. Therefore, future research efforts are much needed in order to pursue better achievements also in the management of severe non-type 2 asthma.

## Figures and Tables

**Figure 1 biomedicines-10-01064-f001:**
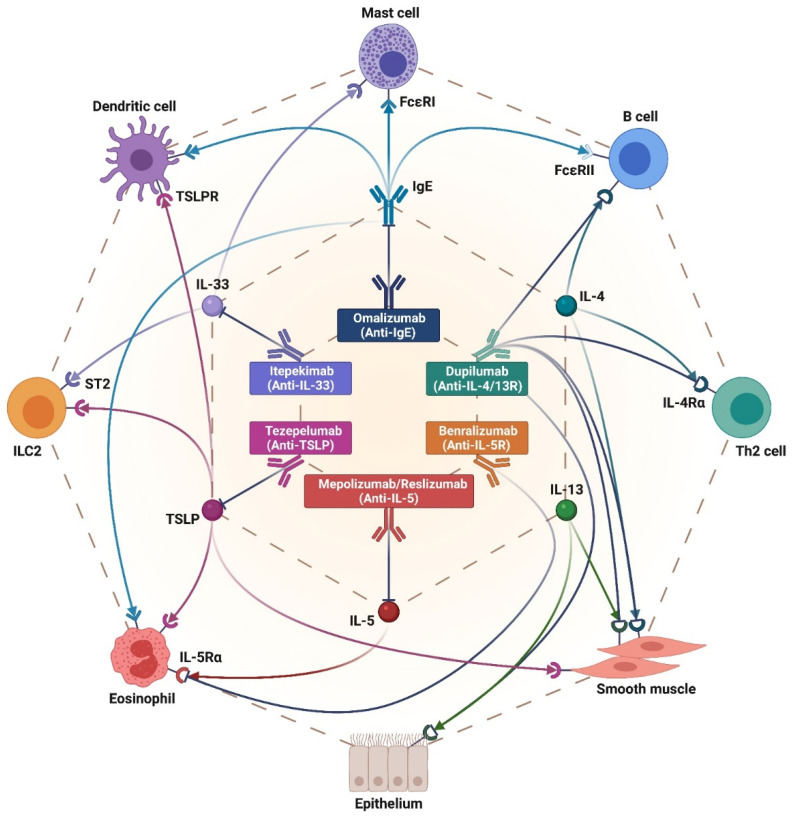
Molecular mechanisms and cellular targets of biological therapies for severe type 2 asthma. Omalizumab, mepolizumab/reslizumab, benralizumab, dupilumab, tezepelumab and itepekimab bind to and effectively inhibit the functions of IgE, IL-5, IL-5 receptor, IL-4/IL-13 receptors, TSLP and IL-33, respectively. Through these mechanisms of action, the above monoclonal antibodies suppress the bioactivities of most airway immune/inflammatory and structural cells involved in the pathogenesis of severe type 2 asthma. This original figure was created by the authors using “BioRender.com” (https://biorender.com, accessed on 9 April 2022).

**Table 1 biomedicines-10-01064-t001:** Biomarker-guided choice of biological treatments for severe asthma.

Biomarker	Endotype	Biological Therapy
High serum IgE concentration	Allergic asthma	Omalizumab
High blood eosinophil count	Eosinophilic asthma	Mepolizumab, Reslizumab, Benralizumab
High FeNO levels	Type 2 asthma	Dupilumab

## Data Availability

Not applicable.

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
