# Peer review of "Novel Biological Therapies for Severe Asthma Endotypes"

_biomedicines, 2022, doi:10.3390/biomedicines10051064_

Round 1

Reviewer 1 Report

The manuscript entitled “ NOVEL BIOLOGICAL THERAPIES FOR SEVERE ASTHMA ENDOTYPES” focused on the pathobiology of severe asthma and current and perspective biological therapies of severe asthma. The authors reported a figure for the description of molecular mechanisms and cellular targets of biological therapies for severe type 2 asthma. Although the manuscript was well structured, there were some that should be revised.

  1. T2 -high or T2-low asthma was too abstract for a reader. Please present the supplementary on page 2.
  2. Please reported the illustration for figure 1.
  3. Please reported a website reference for “BioRender.com”
  4. Too many abbreviations were presented for the first mention throughout the manuscript, such as SIRIUS, MUSCA, and ZENYATTA etc.
  5. The therapies for severe asthma were expounded. However, the authors didn’t introduce the biomarkers for the endotypes. It is suggested with a Table for the comparison.
  6. Please revise FEF25-75 as FEF25-75%.

Author Response

Reviewer 1

  1. T2-high or T2-low asthma was too abstract for a reader. Please present the supplementary on page 2.

The terms “T2-high” and “T2-low” have been replaced by “type 2” and “non-type 2” throughout the revised text.

  1. Please report the illustration for figure 1.

The illustration for Figure 1 has been reported in page 3 of the revised manuscript.

  1. Please report a website reference for “BioRender.com”

The website reference for “BioRender.com” has been included in the legend of Figure 1 (page 3 of the revised manuscript).

  1. Too many abbreviations were presented for the first mention throughout the manuscript, such as SIRIUS, MUSCA, and ZENYATTA etc.

The abbreviations of the names referring to several randomized clinical trials have been deleted throughout the revised text.

  1. The therapies for severe asthma were expounded. However, the authors didn’t introduce the biomarkers for the endotypes. It is suggested with a Table for the comparison.

A table referring to the biomarker-guided choice of the biological treatments of severe asthma, according to the various endotypes, has been included (page 11 of the revised manuscript).

  1. Please revise FEF25-75 as FEF25-75%.

FEF25-75 has been replaced by FEF25-75% (pages 7 and 8 of the revised manuscript).

Reviewer 2 Report

To the Authors:

The purpose of this manuscript was to conduct an updated narrative review on severe asthma. Plausible mechanisms and biologic treatments were described in detail and in relation to severe asthma pathophysiology.

Asthma remains to be the most common allergic disease affecting both children and adults of all ages. Indeed, the severe asthma phenotype is associated with substantial patient morbidity and poor quality of life.  More research and targeted therapies are desperately needed to alleviate symptoms and decrease the risk of future exacerbations.

The manuscript is well-written, comprehensible and the topic has been well-researched and is supported by relevant and latest data.  The authors are to be commended. Well done!

Author Response

Reviewer 2

The manuscript is well-written, comprehensible and the topic has been well-researched and is supported by relevant and latest data. The authors are to be commended. Well done!

We would like to thank very much this reviewer for his/her kind appreciation of our manuscript.